# Molecular Spectroscopic Markers of DNA Damage

**DOI:** 10.3390/molecules25030561

**Published:** 2020-01-28

**Authors:** Kamila Sofińska, Natalia Wilkosz, Marek Szymoński, Ewelina Lipiec

**Affiliations:** M. Smoluchowski Institute of Physics, Jagiellonian University, Łojasiewicza 11, 30-348 Kraków, Poland; kamila.sofinska@uj.edu.pl (K.S.); natalia.szydlowska@uj.edu.pl (N.W.); or ufszymon@cyf-kr.edu.pl (M.S.)

**Keywords:** DNA damage, double strand breaks, DSB, single strand breaks, SSB, DNA lesions, DNA damage spectroscopic markers

## Abstract

Every cell in a living organism is constantly exposed to physical and chemical factors which damage the molecular structure of proteins, lipids, and nucleic acids. Cellular DNA lesions are the most dangerous because the genetic information, critical for the identity and function of each eukaryotic cell, is stored in the DNA. In this review, we describe spectroscopic markers of DNA damage, which can be detected by infrared, Raman, surface-enhanced Raman, and tip-enhanced Raman spectroscopies, using data acquired from DNA solutions and mammalian cells. Various physical and chemical DNA damaging factors are taken into consideration, including ionizing and non-ionizing radiation, chemicals, and chemotherapeutic compounds. All major spectral markers of DNA damage are presented in several tables, to give the reader a possibility of fast identification of the spectral signature related to a particular type of DNA damage.

## 1. Introduction

The human body contains ~10^13^ cells, and each of these cells receives tens of thousands of DNA lesions per day [1]. Several types of DNA damage, such as simple mutations, deamination, missing bases, chemical modification of bases, pyrimidine dimer formation, single strand breaks (SSBs), and double strand breaks (DSBs), are induced (Figure 1). DSB is the most dangerous type of DNA lesion, as it can lead to genetic instability. Determination of disorders in DNA chemical structure is crucial in medical diagnostics [2]. Unrepaired or incorrectly repaired DSBs may block genome replication and transcription, thereby leading to mutations or genome aberrations that threaten the viability of the cell or organism [1]. Therefore, the repair of DSBs is critical to the maintenance of genomic integrity in all forms of life. For self-defense, organisms have developed a few DSB repair mechanisms, of which non-homologous end-joining (NHEJ) and homologous recombination (HR) are the most important [3,4,5].

DNA conformation plays a significant role in the susceptibility to DNA damage and repair processes [6,7]. The double helical structure of DNA, which is stabilized by long-range intra- and inter-backbone forces, including hydrogen bonds between base pairs and stacking interactions between neighboring bases, may exist in different conformations. B-DNA and A-DNA are the main biologically active conformations of the DNA molecule [6]. DNA, in both conformations, has a right-handed double helical structure with major and minor grooves. A-DNA is characterized by a slight increase in the number of base pairs (bp) per turn (a smaller twist angle), in comparison to B-DNA. A smaller rise per bp makes A-DNA 20–25% shorter than B-DNA. A-DNA is broader and more compressed along its axis than B-DNA, and the sizes of the minor and major groves (see Figure 2) are slightly different in the two forms [6]. This property determines the accessibility of other macromolecules to DNA binding sites. The A-form is induced locally in some protein–DNA complexes. It is also shown that damaged DNA subsequently undergoes conformational transition, which promotes its interaction with repair proteins [7]. In addition to the molecule conformation, the structure in general defines the susceptibility to DNA damage formation. Therefore, studies of DNA damage should involve various DNA forms including double helical DNA as well as the physiologically relevant four-stranded DNA structures like G-quadruplexes or i-Motifs (iM) [8,9,10].

DSB formation is directly associated with single strand cleavage events of DNA. The bonds in the DNA backbone (including phosphodiester bonds) may break in one strand of the DNA helix. This could be caused by chemicals like peroxides, enzymes like DNases, and by free radicals, ionizing radiation (protons, heavy ions, X- or gamma rays, electrons), and chemotherapeutic drugs (Figure 1) [11,12]. SSBs are more common than DSBs. However, the damaging factors listed above may cause phosphodiester bond breaks in both strands, which may not be directly opposite to each other. This also leads to DSBs. It is estimated that, for *Homo sapiens*, about 1% of SSBs leads to approximately 50 endogenous DSBs per cell cycle, which means one DSB per 10^8^ bp [13,14]. Some sites on the DNA are more susceptible to damage compared to others. These are called hot-spots. Several possibilities exist for the formation of DSBs, following exposure to one particular damaging factor [15]. The chemical structure of the DSB is determined by the susceptibility of a particular bond (energy and chemical environment) to interact with the damaging factors [1]. In Figure 2, the possible local conformational transition of DNA from the B- to the A-form, upon C–O bond breakage in the DNA backbone, is presented. DSBs can also arise during the replication process, when the replication fork collapses due to encounter with a damaged base or SSB [16]. This type of double-stranded DNA lesion is known as replication fork collapse-induced DSB.

In general, DSBs are more dangerous to genome integrity, and much more attention is required to understand their consequences. However, SSBs can cause serious health disorders as well. It has been proven in a mouse model that the accumulation of SSBs in cardiomyocytes leads to cardiac inflammation in pressure overload-induced heart failure, due to the persistent DNA damage response [17]. To repair SSBs, the base excision repair (BER) mechanism is incorporated during the cell cycle to repair damaged bases. The BER pathway is mainly activated in response to factors like oxidation, hydrolytic deamination, alkylation, ionizing radiation, intracellular metabolites, or ROS due to UV radiation [18,19].

Two principal mechanisms are used for DSB repair: non-homologous end-joining (NHEJ) [20] and homologous recombination (HR) [21]. In NHEJ, DNA ligase IV plays a central role. Initially DSBs are recognized by the Ku protein (Ku-dependent pathways), which then binds and activates the protein kinase DNA catalytic subunit, PKcs, leading to the recruitment and activation of end-processing enzymes, polymerases and DNA ligase IV. The DNA ligase IV complex binds specifically to the ends of duplex DNA molecules, and acts as a bridging factor, linking together duplex DNA molecules with complementary, but non-ligatable, ends. The complex of these three proteins may bind to DNA in vitro, which has been directly observed using AFM [20]. The crystal structure of this complex has also been reported [22]. However, its direct influence on molecular changes in DNA, such as possible conformational transition upon protein complex binding, was never experimentally observed. NHEJ does not require a homologous template, in contrast to the homologous recombination (HR) repair mechanism [23,24,25]. HR requires an undamaged, homologous DNA template for the reconstruction of the incorrect sequence [16,26].

Another example of DNA lesions is the base mismatch. Mismatched nucleotides, G–T or C–A pairs, may be formed due to polymerase disincorporation errors during DNA replication, or due to an incorrect process of physical or chemical DNA damage repair. Errors occur in the non-methylated or daughter strand. The repair mechanism for base mismatch is known as DNA mismatch repair (MMR). The enzymology of MMR has been established. However, many fundamental aspects of the mechanism of MMR remain elusive, such as the structure, composition, and orientation of protein complexes involved in this type of DNA repair [27]. MMR is initiated by the binding of the protein dimer, MutS, to a mismatch, or to a small insertion–deletion loop [28]. MutS then associates with the protein MutL. A complex of these two proteins activates the MutH endonuclease in an ATP-dependent manner. MutH causes an incision in the unmethylated strand, at a d (GATC) site, which can occur at a distance of 1000 bp or more from the mismatch, thus producing a break in the strand at either the 3′ or 5′ end of the mismatch. Subsequently, the unmethylated strand is cleaved at a point beyond the mismatch, and the strand is removed. DNA ligase resynthesizes the excised DNA segment, restoring covalent continuity in the repaired strand [29,30,31]. MutS-mediated loops in DNA have been observed using electron microscopy and AFM [28,29,30,31]. However, both methods did not allow for the investigation of the chemical structure and conformation of DNA.

Spectroscopic techniques are powerful tools for detecting and studying DNA lesions. Their chemical sensitivity makes them the most accurate tool for the detection of even small structural changes [32]. Moreover, techniques based on the interaction of electromagnetic field with chemical bonds enables us to receive the fingerprint spectra of the investigated molecules, allowing for the simultaneous detection of various functional groups typical for the investigated sample. Infrared spectroscopy relies on selective absorption of infrared light by a sample, which is possible because the energies of vibrations and oscillations of the functional groups in molecules correspond to the range of the infrared region. Depending on the absorbed wavelength, from the infrared spectral range, there are far, mid, and near infrared spectroscopies. Each can potentially be applied in studies of DNA damage formation and repair. However, mid infrared spectroscopy is more commonly used in studies of nucleic acids because vibrations of DNA marker bands from DNA backbone, such as phosphate motions, as well as vibrations of DNA bases are in the mid infrared spectral range. These bands can also be observed by Raman spectroscopy. In contrast to IR, Raman spectroscopy is based on inelastic scattering of monochromatic light. The interaction between laser light and molecular vibrations results in the energy of the laser light being shifted up or down. The shift in energy provides information about the vibrational modes in the sample. IR and Raman spectroscopies yield similar information on complex, biological samples such as DNA. However, due to high absorption of the infrared light by water molecules it is more convenient to use Raman spectroscopy. From the other side, some biological molecules are highly fluorescent and relatively weak Raman bands could be overlapped by high fluorescence background. In such a case an application of IR spectroscopy instead of Raman is recommended. Both Raman and IR are sensitive for the conformational changes and modifications of DNA structure.

This review aims to collect and describe spectroscopic markers specific for DNA damage, including SSBs and DSBs, inter- and intra-strand crosslinks, and markers of the conformational transition. This paper is divided into three main parts. The first part focuses on studying DNA damage using traditional spectroscopic methods like Raman and infrared spectroscopy (IR). In the second part, the investigation of DNA lesions with nanospectroscopic methods such as surface enhanced Raman spectroscopy (SERS) and tip-enhanced Raman spectroscopy (TERS) is presented. The third part of this review describes the utilization of confocal Raman and IR spectroscopy for the investigation of cellular response to damaging factors. This part involves a description of spectroscopic markers, found in the spectra acquired from living cells and isolated cellular nuclei.

## 2. DNA Lesions Studied by Spectroscopic Methods

In radiation therapy, the DNA structure is damaged by ionizing radiation in several ways. This includes the removal of large or small pieces of the DNA strand, removal or modification of a base or sugar residue, un-pairing of the base pairs, rupturing of the sugar–phosphate backbone, or DNA cross-linking to the same or different strain/molecule. However, the DNA molecule copes with these damages using various types of enzymes. Additionally, there are serious lesions and double-strand ruptures which are difficult to repair, thereby resulting in cell death. DNA molecules are very sensitive to ionizing radiation [33,34,35,36]. Zhou et al. investigated the influence of ultraviolet radiation (UVR) on calf thymus DNA, in an aqueous solution, following 9, 20 and 40 min of exposure to UVR, using Raman spectroscopy [37]. Experimental results showed that a short exposure time was sufficient to have a strong effect on DNA. Exposure to UVR for 9 min caused a change in the intensity of the DNA band at 1094 cm^−1^. Molecular conformation was also shown to change from B-DNA to A-DNA, which was related to a shift in the asymmetric stretching (O–P–O) of the phosphate bond from 1243 cm^−1^ to 1245 cm^−1^. UVR also had an influence on nitrogen bases, some of which were severely damaged. Synytsya et al. applied Raman spectroscopy to evaluate specific dose-dependent alterations in the double stranded DNA (dsDNA) molecules of calf thymus, following proton and γ-irradiation (0.5–50 Gy) in dilute aqueous solutions of DNA [13]. A decrease in the intensity of the ribose deformational vibration shoulders at 1460 and 1418 cm^−1^, as well as the appearance of new features of terminal phosphates at 1077–1087 cm^−1^, and 982–985 cm^−1^, can be used as markers of SSBs or DSBs in dsDNA, arising due to irradiation [38]. Sailer et al. studied structural changes in dsDNA in aqueous solution, induced by γ-radiation, using Fourier–Transform–Raman spectroscopy. It was shown that the increase in the Raman band intensity was associated with dT, dA, dG, dC, revealing unstacking of these bases. The same effect was observed in Treffer’s study, with the appearance of a band at 1335 cm^−1^ (dA, dG), and ν(C–N) at 1306 cm^−1^ in C, A, G [39]. The unstacking of these bases, caused by proton exposure, was confirmed by the shift of the ν(C=O)—base stacking mode from 1666 cm^−1^ to 1671 cm^−1^ [38]. Observable changes in ν(O–P–O) at 1024 cm^−1^, 1190 cm^−1^, 1080 cm^−1^ and ν(C–O) at 1060 cm^−1^, and at 880 cm^−1^ in the DNA backbone band intensity, were associated with strand breaks and structural changes in the sugar moiety, respectively [38]. Auner et al. studied whether DNA DSBs caused changes in Raman-detected bands, as a result of changes in the vibrational group frequencies in the DNA backbone [40]. They used a chemotherapeutic drug, Bleocin™, for inducing DNA damage [41,42,43]. The Raman bands at 880 cm^−1^, 1044 cm^−1^, 1084 cm^−1^, and 1456 cm^−1^ were significantly greater in the Bleocin™-treated sample, in comparison to the control. In an earlier study, isolated DNA showed Raman bands with intensities at 814 cm^−1^, 783 cm^−1^, 1462 cm^−1^, and 1489 cm^−1^ [44]. Applying Raman spectroscopy, Benevides et al. observed the difference between circular and linear plasmid DNA, using the signature bands at 1456 cm^−1^ and 790 cm^−1^, in a very short 222 bp plasmid [45]. Lipiec et al. studied plasmid DNA affected by UV-C irradiation. All the Raman bands characteristic for DNA damage are shown in Table 1 [15]. The band assignment for purines and pyrimidines was carefully verified with recent literature data, which can be find in an article by Beć and co-workers [46]. Ghosh et al. studied normal human genomic and mutated DNA from K562 leukemic cells, using three different techniques: isothermal titration calorimetry (ITC), dynamic light scattering (DLS), and Raman spectroscopy. They showed the interaction between anticancer drugs (adriamycin and daunomycin) and DNA. The Raman results indicated that the mutated DNA remained partially transformed to the A-DNA form, owing to mutations, while the genomic DNA changed its conformation upon the interaction with the drug [47]. Wood et al. studied the interaction between few platinum drugs and DNA. They found that DNA treated with *trans*-[*N*,*N′*-bis(2,3,5,6-tetrafluorophenyl)ethane-1,2-diaminato(1-)](2,3,4,5,6-pentafluorobenzoato) (pyridine)platinum(II) (PFB) and *trans*-[*N*,*N′*-bis(2,3,5,6-tetrafluorophenyl)ethane-1,2-diaminato(1-)] (2,4,6-trimethylbenzoato)(pyridine)platinum(II) (TMB) transformed from the B- to the A-DNA form irreversibly during dehydration, and did not return back to the B-DNA conformation upon rehydration. The control DNA also gets transformed from the B- to the A-DNA form during dehydration. However, this phenomenon is reversible [48]. Using quasi-elastic neutron scattering (QENS) techniques coupled with synchrotron-based extended X-ray absorption fine structure (SR-EXAFS), and Fourier-transform infrared spectroscopy-attenuated total reflectance (SR-FTIR-ATR), Batista de Carvalho et al. studied the effect of Pt/Pd drugs on the structure of DNA. Far- and mid-infrared measurements allowed for confirming changes in DNA conformation owing to interactions with a chemotherapeutic drug [49].

## 3. Nanospectroscopy of DNA Damage

Conventional spectroscopic methods can be destructive, or show insufficient sensitivity for fine biological samples. For example, the incident frequency for Raman scattering is approximately 1 for 10^10^ photons, which implies low sensitivity, thereby reducing the range of applications [77]. To overcome these limitations, the enhanced effect of Raman scattering on metallic substrates or scanning probe microscopy (SPM) probes, has been explored. The effect of surface enhancement of Raman signal near metallic nanoparticles or nano-patterned metallic surfaces is known as the Surface-enhanced Raman spectroscopy (SERS) [77,78]. SERS offers the possibility for ultrasensitive detection of the rotational and vibrational spectra within biological materials, even for low analyte concentrations [77,79]. Another way to strengthen the weak Raman signal is by using TERS (tip-enhanced Raman scattering), where electromagnetic field enhancement on the SPM tip apex takes place with the 10^6^ factor [80]. TERS combines the spatial resolution (in nm) of SPM techniques like the atomic force microscopy (AFM), or scanning tunneling microscopy (STM), and the chemical selectivity of Raman spectroscopy [81]. A metal nanostructure deposited on the apex of an AFM tip (AFM-TERS), or STM tip apex itself (STM-TERS), modifies the electromagnetic field of the incident laser light, and enhances Raman scattering from the small amount of sample close to the tip. This is due to a near-field enhancement effect caused by surface plasmons, generated at the laser–irradiated tip or apex of a metal, or metallized SPM probe [81,82]. In other words, the tip in TERS operates as a nano-antenna, converting the electric field of the incident laser beam into localized energy [83], thereby improving the spatial resolution of TERS down to few nanometers in vacuum and low temperature conditions [84,85], and altering the near field from the sample to a far field, accessible to the objective, and as a result, increasing the sensitivity up to a single molecule [84].

TERS, as well as SERS, have been exploited to study DNA lesions. The following section focuses on describing examples for utilizing these non-destructive, most sensitive analytical techniques to investigate the chemical structure of individual DNA strands during exposure to damaging factors.

### 2.1. SERS

DNA DSBs induced by reactive oxygen species (ROS) have been studied using SERS, by Panikkanvalappil et al. [86]. The authors focused on revealing marker bands of genomic DNA (isolated from human keratinocyte HaCaT cells) exposed to ROS interaction. To understand the physiological H_2_O_2_/UV influence on DNA damage induction, SERS spectra was collected in a time-dependent manner. A 5 nM solution of Ag nanoparticles (NPs) was used as the Raman signal enhancing source. Structural changes and chemical modifications in DNA molecules after ROS attack detected in SERS spectra (Figure 3) are presented in Table 2.

Another example of ROS’ effect on DNA integrity via SERS was described by Yue et al. [95]. SERS spectra (Figure 4) was collected for a mitochondrial sample extracted from the MCF-7 cells (human breast cancer cells) after photodynamic therapy (PDT) treatment. The living cells were treated with indocyanine green (ICG) for 12 h, and subsequently exposed to an 808 nm laser beam (4 W cm^−2^) for 0, 1, 3, and 5 min. The induction of intracellular ROS following PDT treatment was detected using a non-fluorescent dye, DCFH-DA (2,7-dichlorofluorescin diacetate), which was transferred to fluorescent dichlorofluorescein (DCF), upon ROS-mediated oxidation. To obtain the SERS spectra at the organelle level, the extracted mitochondria was mixed with gold nanoparticles (AuNPs; 0.3 mL, 0.1 nM) in the ratio of 1:1, for 3 min. The marker Raman bands indicating DNA damage following PDT treatment were detected in the spectra acquired from the isolated mitochondria, as summarized in Table 2.

SERS technique has also been exploited to reveal marker bands specific for UV induced DNA damage [78]. In this study, Guselnikova et al. have incorporated an artificial neural network (ANN) to analyze the SERS spectra. For the enhancement of the Raman signal, SERS substrates functionalized with oligonucleotides (OND) complimentary to UV-treated DNA OND, were prepared. The SERS gratings were prepared using Su-8 polymer, deposited by spin-coating with further excimer laser patterning, and Au deposition by vacuum sputtering. Thus, prepared plasmon support surface was modified by 4-carboxybenzenediazonium tosylate (ADT-COOH), in order to form covalent bonds with amino-terminated probe oligonucleotides (p-OND), through EDC/sulfo-NHS coupling. Complementary ONDs were exposed to UV-A light. The degree of OND damage was regulated via exposure time. UV treated ONDs were subsequently coupled, through hybridization, with a surface grafted p-OND. The marker Raman bands, which indicate DNA damage following UV–A exposure, were identified by ANN on SERS spectra, as presented in Table 2. The Raman bands which indicate nitrogenous ring stretching and bending, and vibration of amino groups, were connected with chemical transformations of heterocyclic DNA fragments, which results in the formation of a cyclobutane pyrimidine dimer (CPD), and other photoproducts typical for UV induced DSB and SSB.

Ou et al. studied the influence of X-ray on DNA chemical structure using SERS [79,106]. Various radiation doses (6, 10, 15, and 20 Gy) and cell incubation times following radiation (0, 24, 48, and 72 h) were applied, to determine the chemical modifications of DNA following X-ray exposure. For SERS enhancement, DNA isolated from the nasopharyngeal carcinoma cell line (CNE2) was mixed with Ag nanoparticles. Table 2 summarizes Raman bands collected for different radiation doses and different cell incubation times.

SERS studies of DNA damage primarily focus on the interaction of chemotherapeutic drugs with the DNA molecule. The impact of doxorubicin on oligonucleotide structure was studied by Spadavecchia et al., using pegylated gold nanoparticles (PEG–AuNPs) covalently linked to the cysteamine–gold surface via amide bond, as a SERS substrate [100]. SERS band at 1591 cm^−1^ confirms the formation of the DNA-doxorubicin complex. The authors suggest that the decrease in band intensity at 1586 cm^−1^ indicates the doxorubicin interaction sites, C–O–NH, and the phenyl ring group. The SERS bands characteristic for doxorubicin-DNA interaction are presented in Table 2. The doxorubicin interaction with dsDNA was studied in various SERS configurations, using different SERS enhancers [100,107,108,109], which resulted in a shift in the spectral position of characteristic bands, or a change in their intensity. In many cases, the bands corresponding to the in-plane bending of hydroxyl groups at doxorubicin rings, usually found around 1210 and 1244 cm^−1^, were very weak or absent in the SERS spectrum, owing to drug intercalation into dsDNA structure [101,107].

The effect of cisplatin on the DNA SERS spectra was reported by Masetti et al. [103] and earlier by Barhoumi et al. [110]. Cisplatin is known to form covalent adducts with dsDNA, which affects DNA replication and leads to the activation of signal transduction pathways, and finally to cell apoptosis [103,110]. Masetti used a spermine-coated silver nanoparticle (AgNP@Sp) as a SERS enhancer, while Barhoumi utilized Au nanoshell substrates consisting of 120 nm colloidal silica core and a gold shell. The spectral markers for cisplatin-DNA adducts are listed in Table 2.

### 2.2. TERS

The effect of ultraviolet-C (UV-C) on DNA (pUC18 plasmid)’s chemical structure was determined by Lipiec et al. [15] using TERS. This technique revealed that majority of the dsDNA cleavage accidents were associated with C–O bond breakage. The TERS experiments were conducted in upright configuration using silver coated AFM probes and a green laser beam. The Raman bands obtained for UV–C treated DNA using TERS are presented in Table 3.

## 4. DNA Damage at the Cellular Level

Since molecular spectroscopies such as Raman or IR allow for detailed investigation of the chemical structure of cells and cellular organelles, these tools are extremely useful in studying DNA damage induced in cells and their nuclei. Changes in Raman and infrared spectra associated with the influence of physical and chemical damaging factors on DNA, are summarized in Table 4.

Living melanoma cells, and their extracted nuclei, treated with environmentally relevant fluxes of UVR, were investigated by synchrotron radiation–Fourier transform infrared spectroscopy (SR-FTIR), in combination with principal component analysis (PCA). The living cells and extracted cellular nuclei were studied using SR-FTIR, at three different irradiation dosages (130, 1505, 15 052 Jm^−2^ UVR), after either 24 or 48 h post-irradiation. Evident macromolecular changes were detected. The main spectral change observed was a shift in the DNA asymmetric phosphodiester vibration from 1236 cm^−1^ to 1242 cm^−1^ in the case of the exposed cells, and from 1225 cm^−1^ to 1242 cm^−1^ for the irradiated nuclei. The shifts were ascribed to conformational changes in DNA, such as transition from B-like DNA form to A-like DNA. Additionally, a significant decrease in the peak related to asymmetric phosphodiester vibration, was observed. This is a spectral marker for arresting cells in the G1 phase, due to intensive DNA damage repair. Another significant spectral change observed in cells exposed to photons, and nuclei isolated from the irradiated cells, was a decrease in the absorbance of the base stacking mode at 1714 cm^−1^, related to base-pair damages such as purine and pyrimidine dimer formation, along with 6–4 lesions. The spectral changes listed above are presented in Figure 5, which demonstrates the average spectra of control (untreated) cells and their nuclei, in comparison to the spectra acquired from cells and nuclei treated with three doses of UV radiation. Principal Component Analysis confirmed the obtained results. Plots of scores clearly demonstrated a separation in the spectra acquired from irradiated cells, and the nuclei from non-irradiated controls, in response to the applied range of UVR doses. The achieved results demonstrated the utility of SR-FTIR and Raman spectroscopy in the “in situ” probe of DNA damage, in cell nuclei exposed to UVR.

Additionally, DNA damage caused by ionizing radiation from micro-beams and conventional radiation sources were observed in single fixed cells [112,114,115]. The results showed a dose-dependent shift in the asymmetric stretching of the phosphate bonds from 1234 cm^−1^ to 1237 cm^−1^, related to a local disorder in B-DNA conformation, along with a change in the intensity of the symmetric stretching of a phosphate-related peak at 1083 cm^−1^, which indicates chromatin fragmentation. Such fragmentation is the natural consequence of a high number of DNA DSB [112,114,115].

The radiation dose-dependent changes observed in bands associated with the DNA backbone are characteristic of DNA breakage, including SSBs and DSBs [112,117] and changes in the DNA conformation. Pozzi et al. [113] detected changes in the intensity of the symmetric stretching of phosphate bonds at 1080 cm^−1^, in Jurkat cells irradiated with UVB. Observed variations were explained as evidence of DNA breakage. Similar spectral changes were observed in the spectra of prostate cancer cells (Du-145 cell line) irradiated with 2 MeV protons [112]. Gault et al. observed the difference in the absorbance at 1160 cm^−1^ (–C–OH in nucleic acid and carbohydrates), and interpreted them as an indicator of adjustments to the hydrogen bonding structure in DNA. This effect was also detected in gamma-irradiated lymphocytes by Gault et al. [117]. Similar spectral changes were also observed in HaCaT cells treated with gamma rays by Meade et al. [118].

DNA conformational changes, observed as a shift in the asymmetric stretching of phosphate bonds towards lower energy states, from ~1235 cm^−1^ to ~1245 cm^−1^, were previously detected in fixed cells treated with ionizing [118] and non-ionizing radiation [113,125]. Spectral changes related to DNA damage were not only detected in cells directly treated with photons, but also in neighboring cells, demonstrating that SR-FTIR coupled with multivariate data analysis can be successfully applied for studying the bystander effect [116].

Serious perturbations in the DNA backbone structure can be induced by the application of chemical substances, including intercalating agents. Chan et al. analyzed the ATR-FTIR spectra of MDA-MB-231 cells treated with doxorubicin. A significant decrease was observed, in the bands related to the stretching of O–P–O and C–O at 1080 cm^−1^ and 1050 cm^−1^, respectively. Those spectral changes are related to changes in the DNA-phosphate backbone, because of the DNA disintegrating effect of doxorubicin [121].

The intracellular effects of platinum-based anticancer drugs can be successfully studied using molecular spectroscopy [122,126,127,128]. Schirazi et al. have proven that chemotherapeutic agents such as cisplatin interact primarily with the phosphodiester groups and the phosphate backbone of nucleic acids [123]. The cytotoxic response of living nasopharyngeal carcinoma cells to cisplatin treatment was studied by Huang et al. [122]. They analyzed the bands associated with DNA bases and its backbone. All observed spectral changes induced by cisplatin treatment are listed in the Table 4.

## 5. Conclusions

The mechanism of action of DNA damaging factors can be very complex, and sometimes its experimental verification is very challenging. Molecular spectroscopic techniques such as Raman spectroscopy, TERS/SERS, and/or IR spectroscopy, seem to be very efficient in monitoring molecular changes induced in cells and DNA strands by physical or chemical factors. The undoubted advantage of this approach is the simultaneous observation of the molecular modifications of several functional groups in the DNA backbone (1250–970 cm^−1^), and DNA base pairs (1730–1250 cm-1 and 970–600 cm^−1^). Consequently, the formation of the new chemical bonds between DNA strands and various chemicals can be observed. The breakage of existing DNA bonds can also be detected by molecular spectroscopy, and the chemical structure of the strand breaks can be described based on spectral interpretation. Molecular spectroscopy can also be applied for the investigation of the chemical conformation of DNA. Characteristic shifts of bands associated with the asymmetric and symmetric stretching of the phosphate groups in the DNA backbone towards higher wave numbers, are spectral markers of conformational transition from B-DNA to A-DNA, which is a hallmark of the DNA repair mechanism, and its interaction with other molecules, including the repair proteins. The absorbance change in the base stacking mode at ~1714 cm^−1^ is a spectral marker for a base–pair damage including purine-pyrimidine dimer formation, and 6–4 lesions, typically caused by non-ionizing radiation, chemical compounds, and intercalating agents.

## Figures and Tables

**Figure 1 molecules-25-00561-f001:**
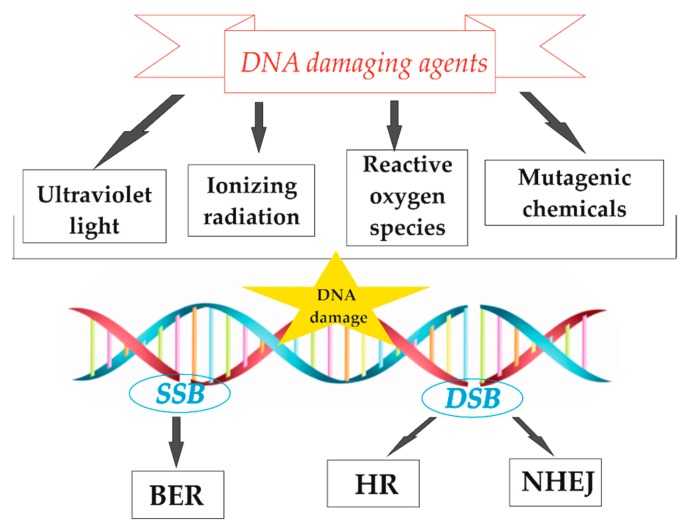
The scheme of exemplary DNA damaging factors, DNA lesion types (SSB—single strand breaks, DSB—double strand breaks), and possible repair mechanisms (BER—base excision repair, HR—homologous recombination, NHEJ—nonhomologous end-joining).

**Figure 2 molecules-25-00561-f002:**
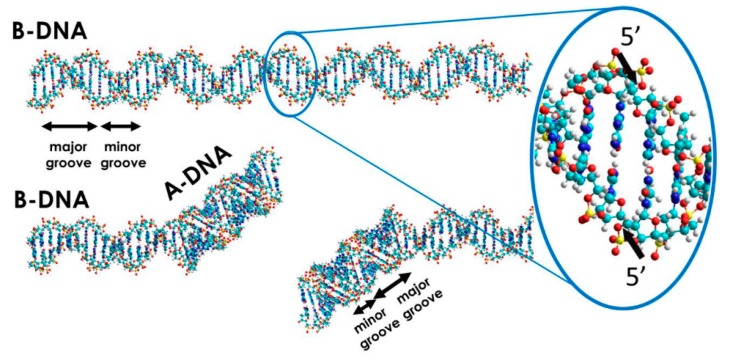
Possible pathways of DSB formation upon C–O bond breakage in the DNA backbone.

**Figure 3 molecules-25-00561-f003:**
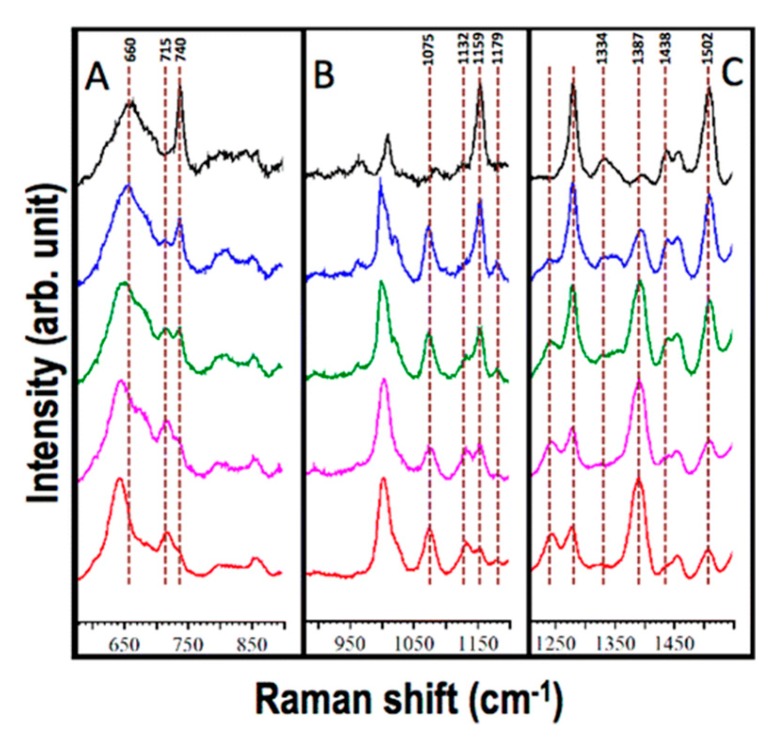
Surface-enhanced Raman spectroscopy (SERS) spectra of the HaCaT cell DNA exposed to H_2_O_2_/UV. Panels A–C indicate regions of interest with characteristic SERS bands of DNA. From top to bottom: SERS spectra collected after 0, 5, 10, 15, and 20 min of exposure to reactive oxygen species (ROS); adapted with permission from [86].

**Figure 4 molecules-25-00561-f004:**
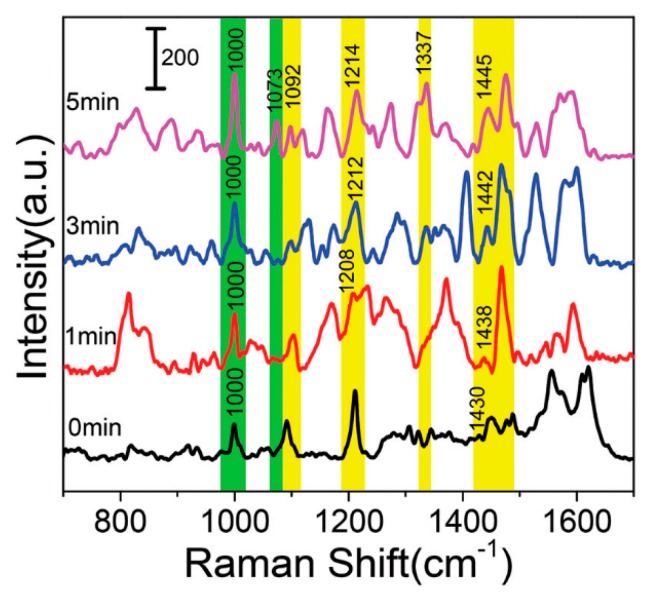
The SERS spectra of mitochondria isolated from MCF-7 cells following photodynamic therapy (PDT) treatment for 0, 1, 3, and 5 min; *λ*_ex_ = 632.8 nm, *t* = 10 s and accumulation = 2 times, adapted with permission from [95].

**Figure 5 molecules-25-00561-f005:**
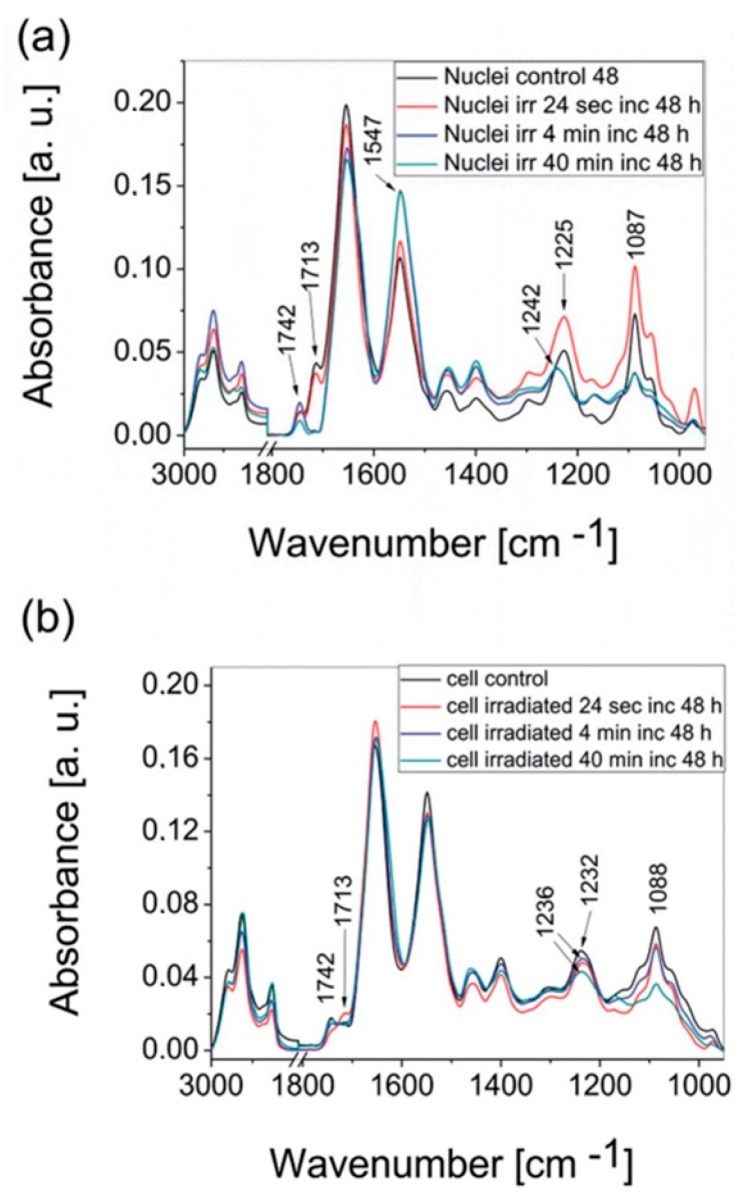
The average spectra collected from isolated COLO-679 cellular nuclei and (**a**) living COLO-679 cells (**b**) control and irradiated cells with exposure to 3 different UV doses, followed by incubation for 48 h; adapted with permission from [112].

**Table 1 molecules-25-00561-t001:** The assignment of DNA damage marker bands.

Damaging Factor	Spectral Change [cm^−1^]	Dynamics	Assignment	Ref.
Ionizing radiation	2898–2891	→	ν(C–H) deoxyribose and T	[39,50,51,52,53]
	1671–1666	→	ν(C=O), δ(NH_2_) dT, dG, dC, C/G/T, ν(C=C str), δ_s_(NH_2_)	
	1579–1576	→	A/C/G/T rings	
	1536–1531	←	in–plane ring str. dC, T	
	1491–1489	→	ν(C=N) in imidiazole, dG, dA,	
	1422–1418	←	ν(C4–C5str) C A, T	
	1339–1335	←	dA, dG, A/G (ring mode)	
	1245–1243	←	ν_asym_(O–P–O)	
	1106–1100	→	ν_sym_(O–P–O)	
	1016–1008	←	ν(N–sugar) A	
	690–670	→	C2’–endo/anti, G ring breathing	
	500–494	→	C/T ring def.	
Proton and γ radiation	1715–1706	→	ν(C6=O)-G	[36,54,55,56,57,58,59,60,61]
	1702–1689	→	ν(C6=O), δ(N1H_2_)-G	
	1684–1683	→	ν(C2=O)-C	
	1670–1664	→↓	ν(C4=O)-T	
	1652–1645	←	ν(C2=O)-T	
	1612–1604	→	ν(C5C6), δ(NH_2_)-A,G	
	1545–1530	←↑	ν(N3C4)-C	
	1515–1508	←	δ(N6H2) + ring-A	
	1488–1482	→↓	ring mode N7-G,A	
	1464–1452	→↓	δ(5′CH_2_)-dRib	
	1427–1418	←↓	δ(2′CH_2_)-dRib	
	1374–1369	→↓	ring mode δ_s_(C5H_3_)-T,A,C	
	1342–1327	→↓	ν(C2N3)-A,G	
	1306–1302	←↓	ring mode-A,C	
	1253–1244	←↓	ring mode-C,T	
	1100–1098	→	ν_s_(PO_2_^−^)	
	1077 app		PO_2_^−^–strand breaks	
	1062–1051	→↓	ν(CO5′)-dRib	
	1013–1003	→↓	T,G,C,dRib	
	985 app		ν(PO_3_^2−^)-strand breaks	
	970–960	←	dRib	
	948–941	→↓	dRib	
	935–927	→↓	dRib	
	918–914	←	ν(CCO)-dRib	
	890–882	←	ν(CO)-dRib-P	
	875 app		dRib-denaturation	
	868–860	→	ν(CO)-dRib-P	
	836–825	←	ν(O–P–O), B-type	
	818–805	←	ν(O–P–O), A-type	
	784–781	→	ring breathing-T,C	
	755–746	←↓	ring breathing-T	
	730–727	→↓	ring breathing-A	
	683–669	→↓	ring breathing-G	
	626–612	→	ring breathing-G,A,dRib	
	599–592	←↓	δ(CCO) + δ(CCC)-dRib; δ(C=O)-C, T	
	572–563	←↓	ring def.-C,T	
	539–535	←↑	ring def.-A	
	500–497	←↓	δ(PO^2−^); ring def.	
	453–450	←↓	ring def.-C	
	424–416	→	ring def.-T,C	
	411–390	←↓	δ(C=O)-T,G	
	334–325	→	ring def.-G,A,T	
Bleomycin	1456	↓	deoxyribose or guanine vibrations	[40]
	1084	↓	ν(PO_2_) bk	
	1044	↓	ν(PO_4_^3−^) bk	
	880	↓	C–C bk vibration	
Ultraviolet C (UVC)	1665–1657	→	δ_s_(NH_2_)	[39,62,63,64,65,66,67,68,69,70,71,72]
	1571–1570	←	in plane ring vibration A/G, δ_s_(NH_2_)	
	1482–1468	→	ν(C=N) pyrimidine, C_2_H_2_ def.	
	1418–1412	→	ν(C6–N1) pyrimidine, ring	
	1336–1332	←	(C–N), ring	
	1303–1293	→	ring	
	1240–1234	←	ν_asym_(O–P–O)	
	1192–1183	←	ν P–(OH)	
	1089–1086	→	ν_sym_(O–P–O)	
	1062–1060	←	ν(N-sugar) A	
	1020–1019	→	ν(C8-N9, N9-H, C8-H) A	
	982–963	←	(CC, CO) T, ribose δr (NH_2_) T	
	916–913	→	δr (NH_2_) A/C/G, deoxyribose	
	784–767	→	T ring breathing	
	680–630	→	G ring breathing	
Adriamycin/daunomycin	1347–1338	→	ring mode(G,A)	[47,73,74]
	1149–1144	←	deoxyribose-phosphate stretching	
	1053–1044	←	ν(C=O)/ν_sym_ PO^2−^	
	962–951	←	deoxyribose	
	889–880	←	deoxyribose/ν(O–P–O)	
	808–800	→	ν(O–P–O) vibration	
	776–763	→	ring breathing (C)	
	701–692	←	ring breathing(G) influence by C2′ endo sugar pucker	
	675–673	←	ring breathing(G) influence by C3′ endo sugar pucker	
	636–633	→	ring mode(T)/T breathing vibration	
Platinum chemotherapeutic drugs	1716–1711	→	base pair carbonyl ν(C=O)	[48,49,75,76]
	1238–1225	→	ν_asym_ PO^2−^	
	1088	↓	ν_sym_ PO_2_	
	1055	↓	ν(C–O) bk	
	968–966	→	ν(C–C) bk	

ν—stretching, ν_sym_—symmetric stretching, ν_asym_—asymmetric stretching, δ—bending, δ_s_—scissoring (in plane bending), δ_r_—rocking (out of plane bending), bk—backbone, def.—deformation, ↓—decrease in intensity, ↑—increase in intensity, ←—shift towards higher wavenumbers in relative to control, →—shift towards lower wavenumbers in relative to control, app—band appearance relative to the control, A—adenine, C—cytosine, G—guanine, T—thymine.

**Table 2 molecules-25-00561-t002:** The assignment of bands in the SERS spectra of DNA treated with various damaging factors.

Damaging Factor	Spectral Change [cm^−1^]	Dynamics	Assignments	Ref.
H_2_O_2_/UV	1502	↓	G, A (oxidation of guanosine by ROS)	[87,88]
	1438	↓	(C5′–H2) def. T	[89]
	1387	↑	A, T, and G ring vibrations	[51,86]
	1334	↓	A ring vibration	
	1179	↑	unpaired T	[63,90]
	1159	↓	A ring vibration	[86]
	1085–1075	→↓	ν_sym_(PO^2−^), bk	[51,86]
	800		ν_sym_(O–P–O)	
	740 spl. to 715 and 738		A, T ring breathing	[51,90]
	660–640	→	G ring breathing (vibration sensitive to orientation relative to the ribose ring)	[91,92,93,94]
ROS inducted during PDT	1445–1430	←↑	ribose; breakage of DNA backbone structure	[95]
	1337	↑	A, cleavage of the double chain	
	1214–1208	←↑	A, cleavage of the double chain	
	1092	↓	ν_sym_(O–P–O); damage of DNA double-helix structure	
UV–A	1660		ν(C=O), δ_s_(NH_2_)	[78]
	1585		ν(C=C, N–C)	
	1480, 1510		ν(N–C)	
	1378		δ(C–H), ν(C–N–C)	
	1270		ν(N–C)	
	1056, 1098		ν_sym_(PO^2−^)	
	904		ν(C–C)	
	802–843		ν_sym_(O–P–O), δ(N–H, C–H)	
	674–703		G ring breathing, δ(C–C, N–C)	
UV-C	1684 dis.		ν(C=O), δ(NH_2_) of T/G/C, base stacking vibration	[50]
	1667–1660	←	δ_s_(NH_2_)	[62]
	1581–1573	←	in plane ring vibration A/G, δ_s_(NH_2_)	[62,63]
	1519–1507	→	ring	[63]
	1484–1478	→	ν(C=N) pyrimidine, (C_2_H_2_) def.	[62,63]
	1440 app.		δ_r_(CH_2_)	[96]
	1402–1400	→	ν(C6–N1) pyrimidine, ring	[62,63]
	1370 app.		δ_s_(CH_2_, CH_3_)	[96]
	1324–1313	→	(C–N), ring	[50,63]
	1324–1311	→	ring	[63]
	1254–1245	←	ν_asym_(O–P–O)	[63]
	1202–1198	→	ν P–(OH)	[64,65]
	1128–1090	→	ν_sym_(O–P–O)	[66]
	1070–1068	←	ν(N–sugar) A	[62]
	1030 dis.		ν(C8–N9, N9–H, C8–H) A	[67]
	930 app.		δ_r_(NH_2_) A/C/G, deoxyribose	[62,67,68]
	880 app.		deoxyribose ring	[69]
	850–841	←	deoxyribose ring	[69]
	786–780	→	T ring breathing	[67,70]
	658–648	→	G ring breathing	[71,72]
X-ray	1577–1563		A, G ring vibration	[51,97,98,99]
	1509–1503		A	
	1420–1414		A, G	
	1336–1333	↑	A, G ring vibration	
	1176–1162		T, C, G,	
	1134–1119		ν(C–N)	
	1095–1086		ν_sym_(O–P–O)	
	893–884		phosphodiester bk, deoxyribose	
	792–783	↑	unpaired T, C, ν_sym_(O–P–O), bk	
	769–767		C, T ring breathing	
	736–730		A ring breathing	
	686–683		G ring breathing	[86,91,94]
Doxorubicin	1642	↓	ν(C=O), hydrogen-bonding to the C=O	[100,101,102]
	1591 app.		complex formation	
	1586–1571	→↓	A; doxorubicin interaction with the N7 position of A, which is accessible for doxorubicin in the DNA structure	
	1467		G, (C8H-N9C8 and C8N7)	
	1372–1312	←↑	T, def. of the hydrogen bond between the NH_2_ group of A and the C40 group of T, δ(ring C20/15 and C16–OH)	
	1318	↑	A; doxorubicin interaction with the N7 position of A	
	1273 app.		_ν(C–O) of ring A of doxorubicin	
	1246, 1214	↓	δ_s_(C–O, C–O–H, C–H) intercalation of rings B and C of doxorubicin within the double helix.	
	1123		ν(C–N)	
Cisplatin	1726	↓	ν(C=O) G	[51,103]
	1665	↑	ν(C=O), (N–H) def. of T/G/C	
	1588	↑	G vibration	
	1485	↓	cisplatin (electrophilic agent) binding to the N7 atom of G	[103,104]
	1333	↑	G vibration	[51,103]
	541 app.		Pt-NH_3_ stretching of cisplatin ligands	[103,105]

ν—stretching, ν_sym_—symmetric stretching, ν_asym_—asymmetric stretching, δ—bending, δ_s_—scissoring (in plane bending), δ_r_—rocking (out of plane bending), bk—backbone, def.—deformation, ↓ decrease in intensity, ↑ increase in intensity, ← shift towards higher wavenumbers relative to control, → shift towards lower wavenumbers relative to control, dis.—band disappearance relative to the control, app.—band appearance relative to the control, spl.—splitting of SERS bands, A—adenine, C—cytosine, G—guanine, T—thymine.

**Table 3 molecules-25-00561-t003:** The assignment of bands in TERS spectra of DNA following UV–C exposure [15].

Damaging Factor	Spectral Change [cm^−1^]	Dynamics	Assignments	Ref.
UV-C	1702 dis.		ν(C=O), δ(NH_2_) of T/G/C, base stacking vibration	[50]
	1653–1651	←	δ_s_(NH_2_)	[62]
	1578–1571	→	in plane ring vibration A/G, δ_s_(NH_2_)	[62,63]
	1515–1512	←	Ring	[63]
	1483–1472	→	ν(C=N) pyrimidine, (C_2_H_2_) def.	[62,63]
	1440 app.		δ_r_ (CH_2_)	[96]
	1406–1403	→	ν(C6–N1) pyrimidine, ring	[62,63]
	(1365–1350) app.		δ_s_(CH_2_, CH_3_)	[96]
	1328–1317	→	(C–N), ring	[50,63]
	1296−(1313–1296)		ring	[63]
	1237–1234	←	ν_asym_(O–P–O)	[63]
	1190 app.		ν P–(OH)	[64,65]
	1160 app.		δ(P–O–H)	[64,96]
	1115		ν_sym_ (P–O–C), ν_sym_ (O–P–O)	[39,66,111]
	1085–1079	→	ν_sym_(O–P–O)	[66]
	1067–1061	→	ν(N–sugar) A	[62]
	1034 app.		ν(C8–N9, N9–H, C8–H) A	[67]
	980 app.		ν(CC, CO) T, ribose; δ_r_(NH_2_) T	[39]
	930 app.		δ_r_(NH_2_) A/C/G, deoxyribose	[62,67,68]
	870 app.		deoxyribose ring	[69]
	830 app.		deoxyribose ring	[69]
	655 app.		G ring breathing	[71,72]

ν—stretching, ν_sym_—symmetric stretching, ν_asym_—asymmetric stretching, δ—bending, δ_s_—scissoring (in plane bending), δ_r_—rocking (out of plane bending), bk—backbone, def.—deformation, ↓ decrease in intensity, ↑ increase in intensity, ← shift towards higher wavenumbers relative to control, → shift towards lower wavenumbers relative to control, dis.—band disappearance relative to the control, app.—band appearance relative to the control, A—adenine, C—cytosine, G—guanine, T—thymine.

**Table 4 molecules-25-00561-t004:** Influence of chemical and physical damaging factors on Raman and IR spectral markers in cells.

Damaging Factor	Type of Damage	Spectral Change [cm^−1^]	Dynamics	Assignment	Ref.
UV-A + UV-B	base-pair damage such as purine and pyrimidine dimer formation along with 6–4 lesions	1714	↓	base stacking mode	[112]
	DNA conformation change	1245–1230	←	ν_asym_(O–P–O)	
	DNA fragmentation, apoptosis	1080	↓	ν_sym_(O–P–O)	[112,113]
Protons	local rupture of base-paired structures	1714	←	Base stacking mode	[114,115,116]
	hydrogen bonding structure in DNA typical for apoptotic cells, changes in the deoxyribose/ribose structure	1242	↓	ν_asym_(O–P–O)	[117,118,119]
		1157	↓←	–C–OH	
	DNA fragmentation, apoptosis	1080	↓	ν_sym_ (O–P–O)	
	DNA fragmentation, apoptosis	970–963	→	ν(C–C), ν(C–O)	[113]
	SSB, DSB, crosslinks, and deoxyribose damage	970	↓	ribose-phosphate skeletal motions	[115]
γ and X rays	base–pair damage including purine, pyrimidine dimer formation and 6–4 lesions	1713	↓	Base stacking mode	[115,120,121]
	partial structural transition from B-DNA and A-DNA	1240–1220	→	ν_asym_(O–P–O)	
		1080	↓	ν_asym_(O–P–O)	
	DNA fragmentation	1036, 1020			
	DNA damage, fragmentation	791	↓	ν_sym_(O–P–O)	
		784	↓	ring breathing vibrations of DNA base pairs	
Doxorubicin	DNA phosphates backbone changes because of the DNA disintegrating effect of doxorubicin	1085	↓	ν_sym_(O–P–O)	[120,121]
		1050	↓	ν(C–O)	
Platinum chemotherapeutic compounds:cisplatin, carboplatin	phosphodiester bonds breakage and DNA bases	1576	↓	A, G	[122]
	phosphodiester bonds breakage and DNA bases	1523	↓	C	
	the change of the DNA content (not change of the double helix structure)	1338		the polynucleotide chain (DNA–purine bases)	
	nonhydrogen-bonded phosphodiester groups of nucleic acids are bonded to a heavy group in cisplatin	(1223–1221)	←	hydrogen-bonded phosphodiester groups	[123]
	modification of the interchain packing of the DNA	1087	↓	ν_sym_(O–P–O)	
	breakdown of phosphodiester bonds and DNA bases	783	↓→	ν(O–P–O)	
Paclitaxel	DNA condensation, apoptosis	1036, 1020	↑	C–O stretch of carbohydrates convoluted with skeletal trans conformation (C–C)	[124]

ν—stretching, ν_sym_—symmetric stretching, ν_asym_—asymmetric stretching, δ—bending, δ_s_—scissoring (in plane bending), δ_r_—rocking (out of plane bending), δ_w_—wagging, δt—twisting, bk—backbone, def.—deformation, ↓—decrease in intensity, ↑—increase in intensity, ←—shift to higher wavenumbers, →—shift to lower wavenumbers, A—adenine, G—guanine, C—cytosine.

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
