# Peer review of "Molecular Spectroscopic Markers of DNA Damage"

_molecules, 2020, doi:10.3390/molecules25030561_

Round 1
Reviewer 1 Report
This is a generally well-readable, elaborate and qualified review article, which is suitable for publishing.
The article is, however, limited to double-stranded DNA and the authors therefore should add some text about four-stranded DNA structures, namely i-motifs and G-quadruplexes:
i-motifs play a role in functionally important parts of the genome (as promoters of telomeres e.g.). There are few studies of ROS impact on i-motifs, see e.g. Dvorakova et al.: Nucleic Acids Research, 2018, Vol. 46, 1624-1634 or Wang et al.: J. Am. Chem. Soc. 2019, 141, 1970-1979, which should be mentioned in the text. ROS impact on G-quadruplex structures has been studied more often (see e.g. Konvalinova et al.: Biochimie, 2015, Vol. 118, 15-25) than on i-motifs and should be part of a review. Please add some information on G-quadruplexes.I found some typos in the tables:
Page 6, table 1: "Platinum" therapeutic drugs
Page 9, table 2, bottom: "Doxorubicin"
Page 9, table 2, lines 6 and 8: "Pyrimidine"
Page 11, table 3, lines 6 and 8: "Pyrimidine"
Some of the text in the table column “Type of damage” sounds a little bizarre, e.g.
page 12, table 4 “local disruption …” and
page 13, table 4 “breakdown of phosphodiester bonds….”.
Please give a structure to these texts to improve readability of table 4.
Please check and brush up the language, see e.g. lines 194, 246 or 257.
Author Response
The list of answers to the reviewers’ questions and comments
The article is, however, limited to double-stranded DNA and the authors therefore should add some text about four-stranded DNA structures, namely i-motifs and G-quadruplexes:
i-motifs play a role in functionally important parts of the genome (as promoters of telomeres e.g.). There are few studies of ROS impact on i-motifs, see e.g. Dvorakova et al.: Nucleic Acids Research, 2018, Vol. 46, 1624-1634 or Wang et al.: J. Am. Chem. Soc. 2019, 141, 1970-1979, which should be mentioned in the text. ROS impact on G-quadruplex structures has been studied more often (see e.g. Konvalinova et al.: Biochimie, 2015, Vol. 118, 15-25) than on i-motifs and should be part of a review. Please add some information on G-quadruplexes.
We would like to thank for this comment. An application of molecular spectroscopy in studies of DNA damage formation in biologically relevant i-motifs and G-quadruplexes would be undoubtedly very important. However to the best of authors knowledge such studies were not performed yet.
We have emphasized the importance of DNA conformation and structure in susceptibility to DNA damage induction, and suggested that various forms of DNA including i-motifs and G-quadruplexes should be used in studies of DNA damage formation.
Please see page 2 lines 52-55.
Each typo has been corrected. Before the submission professional English editing was performed. The certificate is attached to the letter to the editor.
Reviewer 2 Report
Molecules
Review of the manuscript (review) “Molecular Spectroscopic Markers of DNA Damage” (ID: molecules-694167)
In this manuscript, the Authors provide a comprehensive review on spectroscopic markers of cellular DNA lesion. The review is very well-organized, clear and rich in useful information, supported by well-designed tables. Several carefully prepared figures and schemes further enhance the quality of presentation. There is no similar review published in recent literature. It has potentially broader reach, as its scope falls within the interest of researchers and professionals working in several fields. For these reasons, I feel that this work should become much welcomed by the scientific community and the readership of Molecules journal. Before acceptance, a number of minor improvements should be considered, after which the review should become even more informative for the readership.
The reviewed topic is well-presented in this manuscript. However, I feel that the readership would benefit much if Authors attempted to provide some more critical and comparative discussions. For instance, the major attention is given to Raman spectroscopy (including surface plasmon derivatives) with IR spectroscopy and derivatives coming second. FIR (far-IR) spectroscopy is only mentioned as well. It would be advantageous to compare, at least very briefly, the merits and limits of these (and possibly some others as well) techniques in this field of application. A considerable focus is put in the Introduction to biochemistry of DNA, however, almost none to the molecular spectroscopy. I have mixed feelings about putting “popular science” content, e.g. the first sentence in the Introduction. It sounds good and acts on reader’s imagination, the question is, whether it really belongs to the topic. If Authors decided to reduce some of such sentences, some volume for more useful information could be then freed. A relatively high attention is given to assignments of vibrational bands of nucleic acids. Worth checking is the very recent literature focused strictly on assignments for purines and pyrimidines (J. Phys. Chem. B 2019, 123, 47, 10001-10013). I see no reason not to use the abbreviation IR for “infrared” (“infrared spectroscopy” is used multiple times in the manuscript), especially that “IR” is already used in line 331 in the sense of IR radiation and then “IR spectroscopy” is explicitly used in Conclusions (line 369). Rather than saying that “vibrations shift”, it would be more correct to say that “bands shift” (example: line 315). In a similar manner, it is not correct to imply that a mode has intensity. I am also strongly against the term “absorbance intensity” (should rather be “absorbance value”, or just “absorbance” – which is fully legitimate and clear). I presume Authors wanted to avoid repeating similar words and to make the English more diverse. However, care needs to be taken here. Some minor flaws in English should be checked, for example, Authors strongly overuse commas (e.g. in Abstract “because, the genetic…”, and in numerous other sentences). Those, however, do not defeat the scientific clarity of presentation.
Author Response
The list of answers to the reviewers’ questions and comments
For instance, the major attention is given to Raman spectroscopy (including surface plasmon derivatives) with IR spectroscopy and derivatives coming second. FIR (far-IR) spectroscopy is only mentioned as well. It would be advantageous to compare, at least very briefly, the merits and limits of these (and possibly some others as well) techniques in this field of application. A considerable focus is put in the Introduction to biochemistry of DNA, however, almost none to the molecular spectroscopy.
The introduction has been revised and a paragraph describing molecular spectroscopic techniques was added. Advantages and limitations of IR and Raman spectroscopies were also presented. Please see page 4, lines 115-132.
A relatively high attention is given to assignments of vibrational bands of nucleic acids. Worth checking is the very recent literature focused strictly on assignments for purines and pyrimidines (J. Phys. Chem. B 2019, 123, 47, 10001-10013).
We would like to thank for this advice because the correct bands assignment is in central importance. The manuscript was improved as below (please see page 7 lines 184-186).
…“The bands assignment for purines and pyrimidines was carefully verified with recent literature data, which can be find in an article by Beć and co-workers [76].”…
I see no reason not to use the abbreviation IR for “infrared” (“infrared spectroscopy” is used multiple times in the manuscript), especially that “IR” is already used in line 331 in the sense of IR radiation and then “IR spectroscopy” is explicitly used in Conclusions (line 369).
We have unified the abbreviations: IR–infrared, ionizing radiation – no abbreviation.
Rather than saying that “vibrations shift”, it would be more correct to say that “bands shift” (example: line 315). In a similar manner, it is not correct to imply that a mode has intensity. I am also strongly against the term “absorbance intensity” (should rather be “absorbance value”, or just “absorbance” – which is fully legitimate and clear).
Corrected, please see page 16, line 344, 366 and page 17 line 403.
I presume Authors wanted to avoid repeating similar words and to make the English more diverse. However, care needs to be taken here. Some minor flaws in English should be checked, for example, Authors strongly overuse commas (e.g. in Abstract “because, the genetic…”, and in numerous other sentences). Those, however, do not defeat the scientific clarity of presentation.
Before the submission professional English editing was performed, the copy of the certificate is attached to the letter to the editor.